# A GENERALIZED FRAMEWORK OF SEQUENCE GENERATION WITH APPLICATION TO UNDIRECTED SEQUENCE MODELS

## ABSTRACT

Undirected neural sequence models such as BERT (Devlin et al., 2019) have received renewed interest due to their success on discriminative natural language understanding tasks such as question-answering and natural language inference. The problem of generating sequences directly from these models has received relatively little attention, in part because generating from such models departs significantly from the conventional approach of monotonic generation in directed sequence models. We investigate this problem by first proposing a generalized model of sequence generation that unifies decoding in directed and undirected models. The proposed framework models the process of generation rather than a resulting sequence, and under this framework, we derive various neural sequence models as special cases, such as autoregressive, semi-autoregressive, and refinement-based non-autoregressive models. This unification enables us to adapt decoding algorithms originally developed for directed sequence models to undirected models. We demonstrate this by evaluating various decoding strategies for a cross-lingual masked translation model (Lample and Conneau, 2019). Our experiments show that generation from undirected sequence models, under our framework, is competitive with the state of the art on WMT'14 English-German translation. We also demonstrate that the proposed approach enables constant-time translation with similar performance to linear-time translation from the same model by rescoring hypotheses with an autoregressive model.

## 1 INTRODUCTION

Undirected neural sequence models such as BERT (Devlin et al., 2019) have recently brought significant improvements to a variety of discriminative language modeling tasks such as question-answering and natural language inference. Generation of sequences from such models has received relatively little attention. Unlike directed sequence models, each word often depends on the full left and right context around it in undirected sequence models. Thus, a decoding algorithm for an undirected sequence model must specify both how to select positions and what symbols to place in the selected positions. In this paper we formalize this process of selecting positions and replacing symbols as a generalized framework of sequence generation, and unify decoding from both directed and undirected sequence models under this framework. This framing enables us to study generation on its own, independent from the specific parameterization of the sequence models.

Our proposed unified framework casts sequence generation as a process of determining the length of the sequence, and then repeatedly alternating between selecting sequence positions followed by generation of symbols for those positions. A variety of sequence models can be derived under this framework by appropriately designing the length distribution, position selection distribution, and symbol replacement distribution. Specifically, we derive popular neural decoding algorithms such as monotonic autoregressive, non-autoregressive by iterative refinement and monotonic semi-autoregressive decoding as special cases of the proposed model.

This separation of coordinate selection and symbol replacement allows us to build a diverse set of decoding algorithms agnostic to the parameterization or training procedure of the underlying model. We thus fix the symbol replacement distribution as a variant of BERT and focus on deriving novel

generation procedures for undirected neural sequence models under the proposed framework. We design a coordinate selection distribution using a log-linear model and demonstrate that our model generalizes various fixed-order generation strategies, while also being capable of adapting generation order based on the content of intermediate sequences.

We empirically validate our proposal on machine translation using a translation-variant of BERT called a masked translation model (Lample and Conneau, 2019). We design several generation strategies based on features of intermediate sequence distributions and compare them against the state-of-the-art monotonic autoregressive sequence model (Vaswani et al., 2017) on WMT'14 English-German. Our experiments show that generation from undirected sequence models, under our framework, is competitive with the state of the art, and that adaptive-order generation strategies generate sequences in different ways, including left-to-right, right-to-left and mixtures of these. This suggests the potential for designing and learning a more sophisticated coordinate selection mechanism.

Due to the flexibility in specifying a coordinate selection mechanism, we design constant-time variants of the proposed generation strategies, closely following the experimental setup of Ghazvininejad et al. (2019). Our experiments reveal that we can do constant-time translation with the budget as low as 20 iterations (equivalently, generating a sentence of length 20 in the conventional approach) while achieving similar score to linear-time translation from the same masked translation model. This again confirms the potential of the proposed framework and generation strategies.

## 2 A GENERALIZED FRAMEWORK OF SEQUENCE GENERATION

We propose a generalized framework of probabilistic sequence generation to unify both directed and undirected neural sequence models under a single framework. In this generalized framework, we have a *generation sequence* $G$ of pairs of an *intermediate sequence* $Y^t = (y_1^t, \ldots, y_L^t)$ and the corresponding *coordinate sequence* $Z^t = (z_1^t, \ldots, z_L^t)$, where $y_i^t \in V$, $V$ is a vocabulary, $L$ is a length of a sequence, $T$ is a number of generation steps, and $z_i^t \in \{0, 1\}$. The coordinate sequence indicates which of the current intermediate sequence are to be replaced. That is, each consecutive pairs are related to each other by $y_i^{t+1} = (1 - z_i^{t+1})y_i^t + z_i^{t+1}\tilde{y}_i^{t+1}$, where $\tilde{y}_i^{t+1} \in V$ is a new symbol for the position $i$. This sequence of pairs $G$ describes a procedure in which a final sequence $Y^T$ is created, starting from an empty sequence $Y^1 = (\langle \text{mask} \rangle, \ldots, \langle \text{mask} \rangle)$ and empty coordinate sequence $Z^1 = (0, ..., 0)$. This procedure of sequence generation is probabilistically modelled as

$$p(G|X) = \underbrace{p(L|X)}_{\text{(c) length prediction}} \prod_{t=1}^{T}\prod_{i=1}^{L} \underbrace{p(z_i^{t+1}|Y^{\leq t}, Z^t, X)}_{\text{(a) coordinate selection}} \underbrace{p(y_i^{t+1}|Y^{\leq t}, X)}_{\text{(b) symbol replacement}}{}^{z_i^{t+1}}. \tag{1}$$

We condition the whole process on an input variable $X$ to indicate that the proposed model is applicable to both conditional and unconditional sequence generation. In the latter case, $X = \emptyset$.

We first predict the length $L$ of a target sequence $Y$ according to $p(L|X)$ distribution to which we refer as (c) length prediction. At each generation step $t$, we first select the next coordinates $Z^{t+1}$ for which the corresponding symbols will be replaced according to $p(z_i^{t+1}|Y^{\leq t}, Z^t, X)$, to which we refer as (a) coordinate selection. Once the coordinate sequence is determined, we replace the corresponding symbols according to distribution $p(y_i^{t+1}|Y^{\leq t}, Z^{t+1}, X)$, leading to the next intermediate sequence $Y^{t+1}$. From this sequence generation framework, we recover the sequence distribution $p(Y|X)$ by marginalizing out all the intermediate and coordinate sequences except for the final sequence $Y^T$. In the remainder of this section, we describe several special cases of the proposed framework, which are monotonic autoregressive, non-autoregressive, semi-autoregressive neural sequence models.

### 2.1 SPECIAL CASES

**Monotonic autoregressive neural sequence models** We first consider one extreme case of the generalized sequence generation model, where we replace one symbol at a time, monotonically moving from the left-most position to the right-most. In this case, we define the coordinate selection distribution of the generalized sequence generation model in Eq. (1) (a) as $p(z_{i+1}^{t+1} = 1|Y^{\leq t}, Z^t, X) = \mathbb{1}(z_i^t = 1)$, where $\mathbb{1}(\cdot)$ is an indicator function and $z_1^1 = 1$. This coordinate selection distribution is equivalent to saying that we replace one symbol at a time, shifting from the left-most symbol to the right-most symbol, regardless of the content of intermediate sequences. We then choose the symbol replacement distribution in Eq. (1) (b) to be

$p(y_{i+1}^{t+1}|Y^{\leq t}, X) = p(y_{i+1}^{t+1}|y_1^t, y_2^t, \ldots, y_i^t, X)$, for $z_{i+1}^{t+1} = 1$. Intuitively, we limit the dependency of $y_{i+1}^{t+1}$ only to the symbols to its left in the previous intermediate sequence $y_{<(i+1)}^t$ and the input variable $X$. The length distribution (1) (c) is implicitly defined by considering how often the special token $\langle \text{eos} \rangle$, which indicates the end of a sequence, appears after $L$ generation steps: $p(L|X) \propto \sum_{y_{1:L-1}} \prod_{l=1}^{L-1} p(y_{l+1}^{l+1} = \langle \text{eos} \rangle |y_{\leq l}^{\leq l}, X)$. With these choices, the proposed generalized model reduces to $p(G|X) = \prod_{i=1}^L p(y_i|y_{<i}, X)$ which is a widely-used monotonic autoregressive neural sequence model.

**Non-autoregressive neural sequence modeling by iterative refinement** We next consider the other extreme in which we replace the symbols in all positions at every single generation step (Lee et al., 2018). We design the coordinate selection distribution to be $p(z_i^{t+1} = 1|Y^{\leq t}, Z^t, X) = 1 \;\; \forall i \in \{1, ..., L\}$, implying that we replace the symbols in all the positions. We then choose the symbol replacement distribution to be as it was in Eq. (1) (b). That is, the distribution over the symbols in the position $i$ in a new intermediate sequence $y_i^{t+1}$ is conditioned on the entire current sequence $Y^t$ and the input variable $X$. We do not need to assume any relationship between the number of generation steps $T$ and the length of a sequence $L$ in this case. The length prediction distribution $p(L|X)$ is estimated from training data.

**Semi-autoregressive neural sequence models** Wang et al. (2018) recently proposed a compromise between autoregressive and non-autoregresive sequence models by predicting a chunk of symbols in parallel at a time. This approach can also be put under the proposed generalized model. We first extend the coordinate selection distribution of the autoregressive sequence model into

$$p(z_{k(i+1)+j}^{t+1} = 1|Y^{\leq t}, Z^t, X) = \begin{cases} 1, & \text{if } z_{ki+j}^t = 1, \forall j \in \{0, 1, \ldots, k\} \\ 0, & \text{otherwise}, \end{cases}$$

where $k$ is a *group size*. Similarly we modify the symbol replacement distribution:

$$p(y_{k(i+1)+j}^{t+1}|Y^{\leq t}, X) = p(y_{k(i+1)+j}^{t+1}|y_{<k(i+1)}^t, X), \forall j \in \{0, 1, \ldots, k\},$$

for $z_i^t = 1$. This naturally implies that $T = \lceil L/k \rceil$.

# 3 DECODING FROM MASKED LANGUAGE MODELS

In this section, we give an overview of masked language models like BERT, cast Gibbs sampling under the proposed framework, and then use this connection to design a set of approximate, deterministic decoding algorithms for undirected sequence models.

## 3.1 BERT AS AN UNDIRECTED SEQUENCE MODEL

BERT (Devlin et al., 2019) is a masked language model: It is trained to predict a word given the word's left and right context. Because the model gets the full context, there are no directed dependencies among words, so the model is undirected. The word to be predicted is masked with a special $\langle \text{mask} \rangle$ symbol and the model is trained to predict $p(y_i|y_{<i}, \langle \text{mask} \rangle, y_{>i}, X)$. We refer to this as the *conditional BERT distribution*. This objective was interpreted as a stochastic approximation to the pseudo log-likelihood objective (Besag, 1977) by Wang and Cho (2019). This approach of full-context generation with pseudo log-likelihood maximization for recurrent networks was introduced earlier by Berglund et al. (2015). More recently, Sun et al. (2017) use it for image caption generation.

Recent work (Wang and Cho, 2019; Ghazvininejad et al., 2019) has demonstrated that undirected neural sequence models like BERT can learn complex sequence distributions and generate well-formed sequences. In such models, it is relatively straightforward to collect unbiased samples using, for instance, Gibbs sampling. But due to high variance of Gibbs sampling, the generated sequence is not guaranteed to be high-quality relative to a ground-truth sequence. Finding a good sequence in a deterministic manner is also nontrivial.

A number of papers have explored using pretrained language models like BERT to initialize sequence generation models. Ramachandran et al. (2017), Song et al. (2019), and Lample and Conneau (2019) use a pretrained undirected language model to initialize a conventional monotonic autoregressive sequence model, while Edunov et al. (2019) use a BERT-like model to initialize the lower layers of a

generator, without finetuning. Our work differs from these in that we attempt to directly generate from the pretrained model, rather than using it as a starting point to learn another model.

## 3.2 GIBBS SAMPLING IN THE GENERALIZED SEQUENCE GENERATION MODEL

**Gibbs sampling: uniform coordinate selection** To cast Gibbs sampling into our framework, we first assume that the length prediction distribution $P(L|X)$ is estimated from training data, as is the case in the non-autoregressive neural sequence model. In Gibbs sampling, we often uniformly select a new coordinate at random, which corresponds to $p(z_i^{t+1} = 1|Y^{\leq t}, Z^t, X) = 1/L$ with the constraint that $\sum_{i=1}^{L} z_i^t = 1$. By using the conditional BERT distribution as a symbol replacement distribution, we end up with Gibbs sampling.

**Adaptive Gibbs sampling: non-uniform coordinate selection** Instead of selecting coordinates uniformly at random, we can base selections on the intermediate sequences. We propose a log-linear model with features $\phi_i$ based on the intermediate and coordinate sequences:

$$p(z_i^{t+1} = 1|Y^{\leq t}, Z^t, X) \propto \exp\left\{\frac{1}{\tau}\sum_{i=1}^{L}\alpha_i\phi_i(Y^t, Z^t, X, i)\right\}, \qquad (2)$$

again with the constraint that $\sum_{i=1}^{L} z_i^t = 1$. $\tau > 0$ is a temperature parameter controlling the sharpness of the coordinate selection distribution. A moderately high $\tau$ smooths the coordinate selection distribution and ensures that all the coordinates are replaced in the infinite limit of $T$, making it a valid Gibbs sampler (Levine and Casella, 2006).

We investigate three features $\phi_i$: (1) We compute how peaked the conditional distribution of each position given the symbols in all the other positions is by measuring its *negative entropy*: $\phi_{\text{negent}}(Y^t, Z^t, X, i) = -\mathcal{H}(y_i^{t+1}|y_{<i}^t, \langle\text{mask}\rangle, y_{>i}^t, X)$. In other words, we prefer a position $i$ if we know the change in $i$ has a high potential to alter the joint probability $p(Y|X) = p(y_1, y_2, ..., y_L|X)$. (2) For each position $i$ we measure how unlikely the *current* symbol ($y_i^t$, not $y_i^{t+1}$) is under the *new* conditional distribution: $\phi_{\text{logp}}(Y^t, Z^t, X, i) = -\log p(y_i = y_i^t|y_{<i}^t, \langle\text{mask}\rangle, y_{>i}^t, X)$. Intuitively, we prefer to replace a symbol if it is highly incompatible with the input variable and all the other symbols in the current sequence. (3) We encode a *positional preference* that does not consider the content of intermediate sequences: $\phi_{\text{pos}}(i) = -\log(|t - i| + \epsilon)$, where $\epsilon > 0$ is a small constant scalar to prevent $\log 0$. This feature encodes our preference to generate from left to right if there is no information about the input variable nor of any intermediate sequences.

Unlike the special cases of the proposed generalized model in §2, the coordinate at each generation step is selected based on the intermediate sequences, previous coordinate sequences, and the input variable. We mix the features using scalar coefficients $\alpha_{\text{negent}}$, $\alpha_{\text{logp}}$ and $\alpha_{\text{pos}}$, which are selected or estimated to maximize a target quality measure on validation set.

## 3.3 OPTIMISTIC DECODING AND BEAM SEARCH FROM A MASKED LANGUAGE MODEL

Based on the adaptive Gibbs sampler with the coordinate selection distribution in Eq. (2), we can now design an inference procedure to approximately find the most likely sequence from the sequence distribution $p(Y|X)$ by exploiting the corresponding model of sequence generation. In doing so, a naive approach is to marginalize out the generation procedure $G$ using a Monte Carlo method: $\text{argmax}_Y p(Y|X) = \text{argmax}_{Y^T} \frac{1}{M}\sum_{G^m} p(Y^T|Y^{m,<T}, Z^{m,\leq T}, X)$, where $G^m$ is the $m$-th sample from the sequence generation model. This approach suffers from a high variance and non-deterministic behavior, and is less appropriate for practical use. We instead propose an optimistic decoding approach following equation (1):

$$\underset{\substack{L, Y^1, ..., Y^T \\ Z^1, ..., Z^T}}{\text{argmax}} \log p(L|X) + \sum_{t=1}^{T}\sum_{i=1}^{L}\log p(z_i^{t+1}|Y^{\leq t}, Z^t, X) + z_i^{t+1}\log p(y_i^{t+1}|Y^{\leq t}, X). \qquad (3)$$

The proposed procedure is *optimistic* in that we consider a sequence generated by following the most likely generation path to be highly likely under the sequence distribution obtained by marginalizing out the generation path. This optimism in the criterion more readily admits a deterministic approximation scheme such as greedy and beam search, although it is as intractable to solve this problem as the original problem which required marginalization of the generation path.

| | $b$ | $T$ | Baseline | Decoding from an undirected sequence model | | | |
| | | | Autoregressive | Uniform | Left2Right | Least2Most | Easy-First |
|---|---|---|---|---|---|---|---|
| En→De | 1 | $L$ | 25.33 | 21.01 | 24.27 | 23.08 | 23.73 |
| | 4 | $L$ | 26.84 | 22.16 | 25.15 | 23.81 | 24.13 |
| | 4 | $L^*$ | – | 22.74 | 25.66 | 24.42 | 24.69 |
| | 1 | $2L$ | – | 21.16 | 24.45 | 23.32 | 23.87 |
| | 4 | $2L$ | – | 21.99 | 25.14 | 23.81 | 24.14 |
| De→En | 1 | $L$ | 29.83 | 26.01 | 28.34 | 28.85 | 29.00 |
| | 4 | $L$ | 30.92 | 27.07 | 29.52 | 29.03 | 29.41 |
| | 4 | $L^*$ | – | 28.07 | 30.46 | 29.84 | 30.32 |
| | 1 | $2L$ | – | 26.24 | 28.64 | 28.60 | 29.12 |
| | 4 | $2L$ | – | 26.98 | 29.50 | 29.02 | 29.41 |

Table 1: Results (BLEU↑) on WMT'14 En↔De translation using various decoding algorithms and different settings of beam search width ($b$) and number of iterations ($T$) as a function of sentence length ($L$). For each sentence we use 4 most likely sentence lengths. * denotes rescoring generated hypotheses using autoregressive model instead of proposed model.

**Length-conditioned beam search** To solve this intractable optimization problem, we design a heuristic algorithm, called length-conditioned beam search. Intuitively, given a length $L$, this algorithm performs beam search over the coordinate and intermediate token sequences. At each step $t$ of this iterative algorithm, we start from the hypothesis set $\mathcal{H}^{t-1}$ that contains $K$ generation hypotheses: $\mathcal{H}^{t-1} = \left\{ h_k^{t-1} = ((\hat{Y}_k^1, \ldots, \hat{Y}_k^{t-1}), (\hat{Z}_k^1, \ldots, \hat{Z}_k^{t-1})) \right\}_{k=1}^{K}$. Each generation hypothesis has a score:

$$s(h_k^{t-1}) = \log p(L|X) + \sum_{t'=1}^{t-1} \sum_{i=1}^{L} \log p(\hat{z}_i^{t'}|\hat{Y}_k^{<t'}, \hat{Z}^{t'-1}, X) + \hat{z}_i^{t'} \log p(\hat{y}_i^{t'}|\hat{Y}^{\leq t}, X).$$

For notational simplicity, we drop the time superscript $t$. Each of the $K$ generation hypotheses is first expanded with $K'$ candidate positions according to the coordinate selection distribution:

$$\left\{ \hat{h}_{k,k'} \right\}_{k'=1}^{K'} = \arg \text{top-}K'_{u \in \{1,\ldots,L\}} \underbrace{s(h_k) + \log p(z_{k,u} = 1|\hat{Y}^{<t}, \hat{Z}^{t-1}, X)}_{=s(h_k\|\text{one-hot}(u))}$$

so that we have $K \times K'$ candidates $\left\{ \hat{h}_{k,k'} \right\}$, where each candidate consists of a hypothesis $h_k$ with the position sequence extended by the selected position $u_{k,k'}$ and has a score $s(h_k\|\text{one-hot}(u_{k,k'}))$.[1] We then expand each candidate with the symbol replacement distribution:

$$\left\{ \hat{\hat{h}}_{k,k',k''} \right\}_{k''=1}^{K''} = \arg \text{top-}K''_{v \in V} \underbrace{s(h_k\|\text{one-hot}(u_{k,k'})) + \log p(y_{z_{k,k'}} = v|\hat{Y}^{\leq t}, X)}_{=s(h_{k,k'}\|(\hat{Y}_{<z_{k,k'}}^{t-1}, v, \hat{Y}_{>z_{k,k'}}^{t-1}))}.$$

This results in $K \times K' \times K''$ candidates $\left\{ \hat{\hat{h}}_{k,k',k''} \right\}$, each consisting of hypothesis $h_k$ with intermediate and coordinate sequence respectively extended by $v_{k,k',k''}$ and $u_{k,k'}$. Each hypothesis has a score $s(h_{k,k'}\|(\hat{Y}_{<z_{k,k'}}^{t-1}, v_{k,k',k''}, \hat{Y}_{>z_{k,k'}}^{t-1}))$,[2] which we use to select $K$ candidates to form a new hypothesis set $\mathcal{H}^t = \arg \text{top-}K_{h \in \left\{ \hat{\hat{h}}_{k,k',k''} \right\}_{k,k',k''}} s(h)$.

After iterating for a predefined number $T$ of steps, the algorithm terminates with the final set of $K$ generation hypotheses. We then choose one of them according to a prespecified criterion, such as Eq. (3), and return the final symbol sequence $\hat{Y}^T$.

---

[1] $h_k\|\text{one-hot}(u_{k,k'})$ appends one-hot$(u_{k,k'})$ at the end of the sequence of the coordinate sequences in $h_k$

[2] $h_{k,k'}\|(\hat{Y}_{<z_{k,k'}}^{t-1}, v_{k,k',k''}, \hat{Y}_{>z_{k,k'}}^{t-1})$ denotes creating a new sequence from $\hat{Y}^{t-1}$ by replacing the $z_{k,k'}$-th symbol with $v_{k,k',k''}$, and appending this sequence to the intermediate sequences in $h_{k,k'}$.

## 4 EXPERIMENTAL SETTINGS

**Data and preprocessing** We evaluate our framework on WMT'14 English-German translation. The dataset consists of 4.5M parallel sentence pairs. We preprocess this dataset by tokenizing each sentence using a script from Moses (Koehn et al., 2007) and then segmenting each word into subword units using byte pair encoding (Sennrich et al., 2016) with a joint vocabulary of 60k tokens. We use newstest-2013 and newstest-2014 as validation and test sets respectively.

**Sequence models** We base our models off those of Lample and Conneau (2019). Specifically, we use a Transformer (Vaswani et al., 2017) with 1024 hidden units, 6 layers, 8 heads, and Gaussian error linear units (Hendrycks and Gimpel, 2016). We use a pretrained model[3] trained using a masked language modeling objective (Lample and Conneau, 2019) on 5M monolingual sentences from WMT NewsCrawl 2007-2008. To distinguish between English and German sentences, a special language embedding is added as an additional input to the model.

We adapt the pretrained model to translation by finetuning it with a masked translation objective (Lample and Conneau, 2019). We concatenate parallel English and German sentences, mask out a subset of the tokens in either the English or German sentence, and predict the masked out tokens. We uniformly mask out $0-100\%$ tokens as in Ghazvininejad et al. (2019). Training this way more closely matches the generation setting where the model starts with an input sequence of all masks.

**Baseline model** We compare against a standard Transformer encoder-decoder autoregressive neural sequence model (Vaswani et al., 2017) trained for left-to-right generation and initialized with the same pretrained model. We train a separate autoregressive model to translate an English sentence to a German sentence and vice versa, with the same hyperparameters as our model.

**Training details** We train the models using Adam (Kingma and Ba, 2014) with an inverse square root learning rate schedule, learning rate of $10^{-4}$, $\beta_1 = 0.9$, $\beta_2 = 0.98$, and dropout rate of $0.1$ (Srivastava et al., 2014). Our models are trained on $8$ GPUs with a batch size of $256$ sentences.

**Decoding strategies** We design four generation strategies for the masked translation model based on the log-linear coordinate selection distribution in §2:

1. **Uniform**: $\tau \to \infty$, i.e., sample a position uniformly at random without replacement
2. **Left2Right**: $\alpha_{\text{negent}} = 0$, $\alpha_{\text{logp}} = 0$, $\alpha_{\text{pos}} = 1$
3. **Least2Most** (Ghazvininejad et al., 2019): $\alpha_{\text{negent}} = 0$, $\alpha_{\text{logp}} = 1$, $\alpha_{\text{pos}} = 0$
4. **Easy-First**: $\alpha_{\text{negent}} = 1$, $\alpha_{\text{logp}} = 1$,[4] $\alpha_{\text{pos}} = 0$

We use beam search described in §3.3 with $K'$ fixed to 1, i.e., we consider only one possible position for replacing a symbol per hypothesis each time of generation. We vary $K = K''$ between 1 (greedy) and 4. For each source sentence, we consider four length candidates according to the length distribution estimated from the training pairs, based on early experiments showing that using only four length candidates performs as

|  | Gold | # of length candidates | | | |
|---|---|---|---|---|---|
|  | | 1 | 2 | 3 | 4 |
| En→De | 22.50 | 22.22 | 22.76 | 23.01 | 23.22 |
| De→En | 28.05 | 26.77 | 27.32 | 27.79 | 28.15 |

Table 2: Effect of the number of length candidates considered during decoding on BLEU, measured on the validation set (newstest-2013) using the **easy-first** strategy.

well as using the ground-truth length (see Table 2). Given the four candidate translations, we choose the best one according to the pseudo log-probability of the final sequence (Wang and Cho, 2019). Additionally, we experimented with choosing best translation according to log-probability of the final sequence calculated by an autoregressive neural sequence model.

**Decoding scenarios** We consider two decoding scenarios: linear-time and constant-time decoding. In the linear-time scenario, the number of decoding iterations $T$ grows linearly w.r.t. the length of a target sequence $L$. We test setting $T$ to $L$ and $2L$. In the constant-time scenario, the number of iterations is constant w.r.t. the length of a translation, i.e., $T = O(1)$. At the $t$-th iteration of generation, we replace $o_t$-many symbols, where $o_t$ is either a constant $\lceil L/T \rceil$ or linearly anneals from $L$ to $1$ ($L \to 1$) as done by Ghazvininejad et al. (2019).

---

[3]https://dl.fbaipublicfiles.com/XLM/mlm_ende_1024.pth
[4]We set $\alpha_{\text{logp}} = 0.9$ for De→En based on the validation set performance.

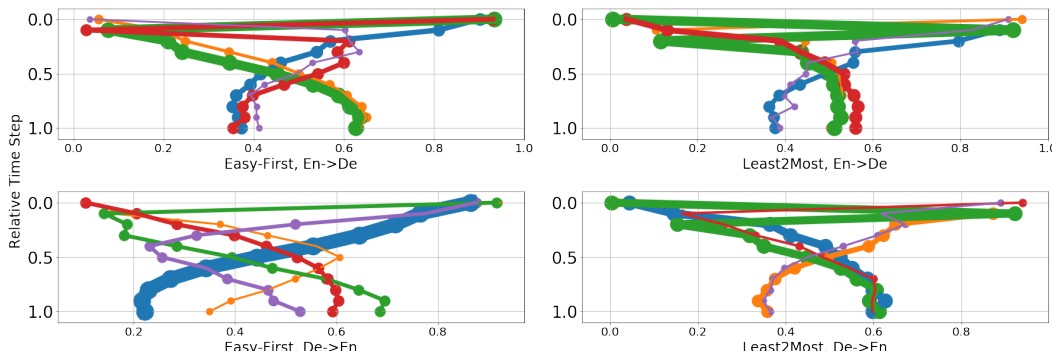

Figure 1: Generation orders given by **easy-first** and **least2most** coordinate selection. We group the orders into five clusters and visualize cluster centers with normalized positions (x-axis) over normalized steps (y-axis). We use greedy search with $L$ iterations on the development set.

## 5 LINEAR-TIME DECODING: RESULT AND ANALYSIS

**Main findings** We present translation quality measured by BLEU (Papineni et al., 2002) in Table 1. We identify a number of important trends. (1) The deterministic coordinate selection strategies (**left2right**, **least2most** and **easy-first**) significantly outperform selecting coordinates uniformly at random, by up to 3 BLEU in both directions. The success of these relatively simple hand-crafted coordinate selection strategies suggest avenues for further improvement for generation from undirected sequence models. (2) The proposed beam search algorithm for undirected sequence models provides an improvement of about 1 BLEU over greedy search, confirming the utility of the proposed framework as a way to move decoding techniques across different paradigms of sequence modeling. (3) Rescoring generated translations with an autoregressive model adds about 1 BLEU across all coordinate selection strategies. Rescoring adds minimal overhead as it is run in parallel. (4) Different generation strategies result in translations of varying qualities depending on the setting. On English-German translation, **left2right** is the best performing strategy, achieving up to 25.66 BLEU. **Easy-first** and **left2right** perform nearly the same in the other direction, achieving up to 30.46 BLEU. (5) We see little improvement in refining a sequence beyond the first pass, though we suspect this may be due to the simplicity of the coordinate selection schemes tested. (6) Lastly, the masked translation model lags behind the more conventional neural autoregressive model, although the difference is within 1 BLEU point when greedy search is used with the autoregressive model and approximately 2 BLEU with beam search. We hypothesize that a difference between train and test settings causes a slight performance drop of masked translation model compared to conventional autoregressive model. In the standard autoregressive case, the model is explicitly trained to generate in left-to-right order, which matches the test time usage. By randomly selecting tokens to mask during training, our undirected sequence model is trained to follow all possible generation orders and to use context from both directions, which is not available when generating left-to-right at test time.

**Adaptive generation order** The **least2most** and **easy-first** generation strategies automatically adapt the generation order based on the intermediate sequences generated. We investigate the resulting generation orders on the development set by presenting each as a 10-dim vector (downsampling as necessary), where each element corresponds to the selected position in the target sequence normalized by sequence length. We cluster these sequences with $k$-means clustering and visualize the clusters centers as curves with thickness proportional to the number of sequences in the cluster in Fig. 1.

In both strategies, we see two major trends. First, many sequences are generated largely monotonically either left-to-right or right-to-left (see, e.g., green, blue and orange clusters in **easy-first**, En→De, and blue, orange, and red clusters in **least2most**, De→En.) Second, another cluster of sequences are generated from outside in, as seen in the red and purple clusters in **easy-first**, En→De, and green, orange, and purple clusters in **least2most**, En→De. We explain these two behaviors by the availability of contextual evidence, or lack thereof. At the beginning of generation, the only two non-mask symbols are the beginning and end of sentence symbols, making it easier to predict a symbol at the beginning or end of the sentence. As more symbols are filled near the boundaries, more evidence is accumulated for the decoding strategy to accurately predict symbols near the center.

| $T$ | $o_t$ | Uniform | Left2Right | Least2Most | Easy-First | Hard-First |
|-----|-------|---------|------------|------------|------------|------------|
| 10 | $L \to 1$ | 22.38 | 22.38 | 27.14 | 22.21 | 26.66 |
| 10 | $L \to 1^*$ | 23.64 | 23.64 | 28.63 | 23.79 | 28.46 |
| 10 | $\lceil L/T \rceil$ | 22.43 | 21.92 | 24.69 | 25.16 | 23.46 |
| 20 | $L \to 1$ | 26.01 | 26.01 | 28.54 | 22.24 | 28.32 |
| 20 | $L \to 1^*$ | 27.28 | 27.28 | 30.13 | 24.55 | 29.82 |
| 20 | $\lceil L/T \rceil$ | 24.69 | 25.94 | 27.01 | 27.49 | 25.56 |

Table 3: Constant-time machine translation on WMT'14 De→En with different settings of the budget ($T$) and number of tokens predicted each iteration ($o_t$). * denotes rescoring generated hypotheses using autoregressive model instead of proposed model.

This process manifests either as monotonic or outside-in generation. We present sample sequences generated using these strategies in Appendix D.

## 6    Constant-Time Decoding: Result and Analysis

The trends in constant-time decoding noticeably differ from those in linear-time decoding. First, the **left2right** strategy significantly lags behind the **least2most** strategy, and the gap is wider (up to 4.7 BLEU) with a tighter budget ($T = 10$). This gap suggests that a better, perhaps learned, coordinate selection scheme could further improve constant-time translation. Second, the **easy-first** strategy is surprisingly the worst in constant-time translation, unlike in linear-time translation. To investigate this degradation, we test another strategy where we flip the signs of the coefficients in the log-linear model. This new **hard-first** strategy works on par with **least2most**, which again confirms that decoding strategies must be selected based on the target tasks and decoding setting.

With a fixed budget of $T = 20$, linearly annealing $K$, and **least2most** decoding, constant-time translation can achieve translation quality comparable to linear-time translation with the same model (30.13 vs. 30.46), and to beam-search translations using the strong neural autoregressive model (30.13 vs 30.92). Even with a tighter budget of 10 iterations (less than half the average sentence length), constant-time translation loses only 1.8 BLEU points (28.63 vs. 30.46), which confirms the finding by Ghazvininejad et al. (2019) and offers new opportunities in advancing constant-time machine translation systems. Compared to other constant-time machine translation approaches, our model outperforms many recent approaches by Gu et al. (2018); Lee et al. (2018); Wang et al. (2019); Ma et al. (2019), while being comparable to Ghazvininejad et al. (2019); Shu et al. (2019). We present full results in Table 4 in Appendix A.

## 7    Conclusion

We present a generalized framework of neural sequence generation that unifies decoding in directed and undirected neural sequence models. Under this framework, we separate position selection and symbol replacement, allowing us to apply a diverse set of generation algorithms, inspired by those for directed neural sequence models, to undirected models such as BERT and its translation variant.

We evaluate these generation strategies on WMT'14 En-De machine translation using a recently proposed masked translation model. Our experiments reveal that undirected neural sequence models achieve performance comparable to conventional, state-of-the-art autoregressive models, given an appropriate choice of decoding strategy. We further show that constant-time translation in these models performs similar to linear-time translation by using one of the proposed generation strategies. Analysis of the generation order automatically determined by these adaptive decoding strategies reveals that most sequences are generated either monotonically or outside-in.

We identify two promising extensions to our work. First, we could have a model learn the coordinate selection distribution from data to maximize translation quality. Doing so would likely result in better quality sequences as well as the discovery of more non-trivial generation orders. Second, we only apply our framework to sequence generation, but we could also apply it to other structured data such as grids (for e.g. images) and arbitrary graphs. Overall, we hope that our generalized framework opens new avenues in developing and understanding generation algorithms for a variety of settings.

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

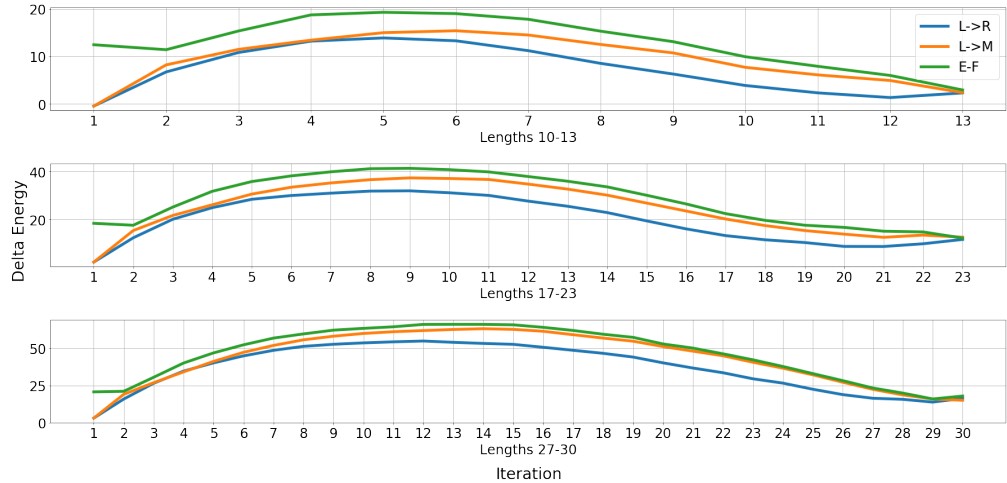

Figure 2: Average difference in energy ↑ between sequences generated by selecting positions uniformly at random versus by different algorithms, over the course of decoding.

## A  COMPARISON WITH OTHER NON-AUTOREGRESSIVE NEURAL MACHINE TRANSLATION APPROACHES

We present the comparison of results of our approach with other constant-time machine translation approaches in Table 4.

## B  NON-MONOTONIC NEURAL SEQUENCE MODELS

The proposed generalized framework subsumes recently proposed variants of non-monotonic generation (Welleck et al., 2019; Stern et al., 2019; Gu et al., 2019). Unlike the other special cases described above, these non-monotonic generation approaches learn not only the symbol replacement distribution but also the coordinate selection distribution, and implicitly the length distribution, from data. Because the length of a sequence is often not decided in advance, the intermediate coordinate sequence $Z^t$ and the coordinate selection distribution are reparameterized to work with relative coordinates rather than absolute coordinates. We do not go into details of these recent algorithms, but we emphasize that all these approaches are special cases of the proposed framework, which further suggests other variants of non-monotonic generation.

## C  ENERGY EVOLUTION OVER GENERATION STEPS

While the results in Table 1 indicate that our decoding algorithms find better generations in terms of BLEU relative to uniform decoding, we verify that the algorithms produce generations that are more likely according to the model. We do so by computing the energy (negative logit) of the sequence of intermediate sentences generated while using an algorithm, and comparing to the average energy of intermediate sentences generated by picking positions uniformly at random. We plot this energy difference over decoding in Figure 2. We additionally plot the evolution of energy of the sequence by different position selection algorithms throughout generation process in Figure 3. Overall, we find that left-to-right, least-to-most, and easy-first do find sentences that are lower energy than the uniform baseline over the entire decoding process. Easy-first produces sentences with the lowest energy, followed by least-to-most, and then left-to-right.

## D  SAMPLE SEQUENCES AND THEIR GENERATION ORDERS

We present sample decoding processes using the easy-first decoding algorithm on De→En with $b = 1, T = L$ in Figures 4, 5, 6, and 7. We highlight examples decoding in right-to-left-to-right-

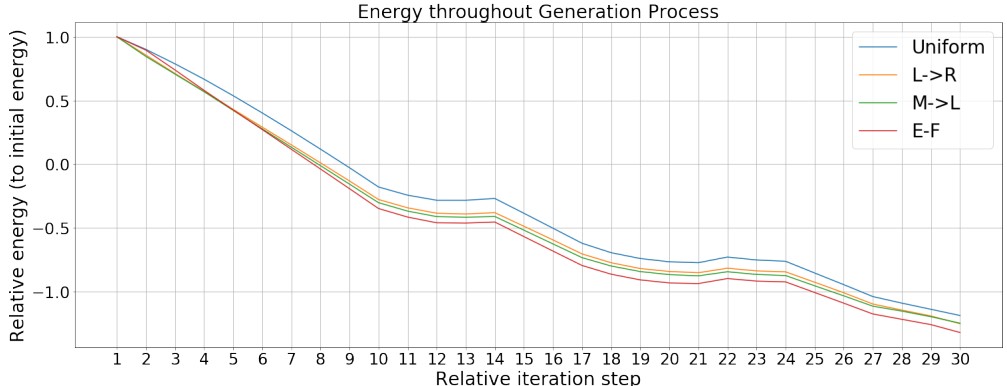

Figure 3: Evolution of the energy of the sequence ↓ over the course of decoding by different position selection algorithms.

to-left order, outside-in, left-to-right, and right-to-left orders, which respectively correspond to the orange, purple, red, and blue clusters from Figure 1, bottom left. These example demonstrate the ability of the easy-first coordinate selection algorithm to adapt the generation order based on the intermediate sequences generated. Even in the cases of largely monotonic generation order (left-to-right and right-to-left), the algorithm has the capacity to make small changes to the generation order as needed.

| Models | WMT2014 | |
| --- | --- | --- |
| | EN-DE | DE-EN |
| AR Transformer-base (Vaswani et al., 2017) | 27.30 | – |
| AR (Gu et al., 2018) | 23.4 | – |
| NAR (+Distill +FT +NPD S=100) | 21.61 | – |
| AR (Lee et al., 2018) | 24.57 | 28.47 |
| Adaptive NAR Model | 16.56 | – |
| Adaptive NAR Model (+Distill) | 21.54 | 25.43 |
| AR (Wang et al., 2019) | 27.3 | 31.29 |
| NAT-REG (+Distill) | 20.65 | 24.77 |
| NAT-REG (+Distill +AR rescoring) | 24.61 | 28.90 |
| AR (Ghazvininejad et al., 2019) | 27.74 | 31.09 |
| CMLM with 4 iterations | 22.25 | – |
| CMLM with 4 iterations (+Distill) | 25.94 | 29.90 |
| CMLM with 10 iterations | 24.61 | – |
| CMLM with 10 iterations (+Distill) | 27.03 | 30.53 |
| AR (Shu et al., 2019) | 26.1 | – |
| Latent-Variable NAR | 11.8 | – |
| Latent-Variable NAR (+Distill) | 22.2 | – |
| Latent-Variable NAR (+Distill +AR Rescoring) | 25.1 | – |
| AR (Ma et al., 2019) | 27.16 | 31.44 |
| FlowSeq-base (+NPD n = 30) | 21.15 | 26.04 |
| FlowSeq-base (+Distill +NPD n = 30) | 23.48 | 28.40 |
| AR (ours) | 26.84 | 30.92 |
| Contant-time 10 iterations | 21.98 | 27.14 |
| Contant-time 10 iterations (+AR Rescoring) | 24.53 | 28.63 |
| Contant-time 20 iterations | 23.92 | 28.54 |
| Contant-time 20 iterations (+AR Rescoring) | 25.69 | 30.13 |

Table 4: BLEU scores on WMT'14 En→De and De→En datasets showing performance of various constant-time machine translation approaches. Each block shows the performance of autoregressive model baseline with their proposed approach. AR denotes autoregressive model. Distill denotes distillation. AR rescoring denotes rescoring of samples with autoregressive model. FT denotes fertility. NPD denotes noisy parallel decoding followed by rescoring with autoregressive model.

| Iteration | Right-to-Left-to-Right-to-Left |
|---|---|
| (source) | Würde es mir je gelingen , an der Universität Oxford ein normales Leben zu führen ? |
| 1 | _ _ _ _ _ _ _ _ _ _ _ _ _ _ _ _ **?** |
| 2 | _ _ _ _ _ _ _ _ _ _ _ _ _ _ **Oxford** ? |
| 3 | _ _ **ever** _ _ _ _ _ _ _ _ _ _ _ Oxford ? |
| 4 | _ **I** ever _ _ _ _ _ _ _ _ _ _ _ Oxford ? |
| 5 | _ I ever _ _ _ _ _ _ _ _ _ _ **of** Oxford ? |
| 6 | **Would** I ever _ _ _ _ _ _ _ _ _ _ of Oxford ? |
| 7 | Would I ever _ _ _ _ _ **normal** _ _ _ _ of Oxford ? |
| 8 | Would I ever _ _ _ _ _ normal _ **at** _ _ of Oxford ? |
| 9 | Would I ever _ _ _ _ _ normal _ at **the** _ of Oxford ? |
| 10 | Would I ever _ _ _ _ _ normal _ at the **University** of Oxford ? |
| 11 | Would I ever _ _ _ _ _ normal **life** at the University of Oxford ? |
| 12 | Would I ever _ _ _ **live** _ normal life at the University of Oxford ? |
| 13 | Would I ever _ _ _ live **a** normal life at the University of Oxford ? |
| 14 | Would I ever _ **able** _ live a normal life at the University of Oxford ? |
| 15 | Would I ever **be** able _ live a normal life at the University of Oxford ? |
| 16 | Would I ever be able **to** live a normal life at the University of Oxford ? |
| (target) | Would I ever be able to lead a normal life at Oxford ? |

Figure 4: Example sentences generated following an right-to-left-to-right-to-left generation order, using the easy-first decoding algorithm on De→En.

| Iteration | Outside-In |
|---|---|
| (source) | Doch ohne zivilgesellschaftliche Organisationen könne eine Demokratie nicht funktionieren . |
| 1 | _ _ _ _ _ _ _ _ _ _ **.** |
| 2 | _ _ _ _ _ _ _ _ **cannot** _ . |
| 3 | _ _ _ _ _ _ _ **democracy** cannot _ . |
| 4 | _ **without** _ _ _ _ _ democracy cannot _ . |
| 5 | _ without _ _ _ _ _ democracy cannot **work** . |
| 6 | **But** without _ _ _ _ _ democracy cannot work . |
| 7 | But without _ _ _ _ **a** democracy cannot work . |
| 8 | But without _ **society** _ _ a democracy cannot work . |
| 9 | But without _ society _ **,** a democracy cannot work . |
| 10 | But without **civil** society _ , a democracy cannot work . |
| 11 | But without civil society **organisations** , a democracy cannot work . |
| (target) | Yet without civil society organisations , a democracy cannot function . |

Figure 5: Example sentences generated following an outside-in generation order, using the easy-first decoding algorithm on De→En.

| Iteration | Left-to-Right |
|---|---|
| (source) | Denken Sie , dass die Medien zu viel vom PSG erwarten ? |
| 1 | _ _ _ _ _ _ _ _ _ _ _ **?** |
| 2 | **Do** _ _ _ _ _ _ _ _ _ _ ? |
| 3 | Do **you** _ _ _ _ _ _ _ _ _ ? |
| 4 | Do you **think** _ _ _ _ _ _ _ _ ? |
| 5 | Do you think _ _ _ _ _ _ **PS** _ ? |
| 6 | Do you think _ _ _ _ _ _ PS **@G** ? |
| 7 | Do you think _ **media** _ _ _ _ PS @G ? |
| 8 | Do you think **the** media _ _ _ _ PS @G ? |
| 9 | Do you think the media **expect** _ _ _ PS @G ? |
| 10 | Do you think the media expect _ **much** _ PS @G ? |
| 11 | Do you think the media expect **too** much _ PS @G ? |
| 12 | Do you think the media expect too much **of** PS @G ? |
| (target) | Do you think the media expect too much of PS @G ? |

Figure 6: Example sentences generated following an left-to-right generation order, using the easy-first decoding algorithm on De→En.

| Iteration | Right-to-Left |
|---|---|
| (source) | Ein weiterer Streitpunkt : die Befugnisse der Armee . |
| 1 | _ _ _ _ _ _ _ _ _ _ . |
| 2 | _ _ _ _ _ _ _ _ _ **army** . |
| 3 | _ _ _ _ _ _ _ **of** _ army . |
| 4 | _ _ _ _ _ _ _ of **the** army . |
| 5 | _ _ _ _ _ _ **powers** of the army . |
| 6 | _ _ _ _ _ **the** powers of the army . |
| 7 | _ _ _ _ **:** the powers of the army . |
| 8 | _ _ **point** : the powers of the army . |
| 9 | _ **contentious** point : the powers of the army . |
| 10 | **Another** contentious point : the powers of the army . |
| (target) | Another issue : the powers conferred on the army . |

Figure 7: Example sentences generated following an right-to-left generation order, using the easy-first decoding algorithm on De→En.

