# OpenReview forum: "A Generalized Framework of Sequence Generation with Application to Undirected Sequence Models"
_ICLR.cc/2020/Conference — Reject_

### Official Review · AnonReviewer3 · 2019-10-23
**Official Blind Review #3**

**Rating:** 3

**Review:**

This paper presents a general framework for sentence generation using a BERT-like model. The authors decompose the problem of sentence generation into two problems. One is selecting the positions at which changes should be made, and the other is actually replacing the current word with a new word. This framework enables them to represent many decoding strategies including that of Ghazvininejas et al. (2019) in a unified manner, and they propose a new decoding strategy that considers the prediction confidence of the current and the new word. The paper also presents a heuristic algorithm for beam search decoding to find the most likely generation path. Their experimental results on the WMT14 English-German dataset suggest that the proposed approach could achieve translation quality comparable to that of the standard autoregressive approach under a constant-time translation setting.

It is nice to see existing decoding strategies represented in a generalized framework, but I was a bit disappointed that the authors do not seem to address the most critical problem in using a BERT-like model for sentence generation, namely, how to find the most likely sentence in a probabilistically sound way. It seems to me that the authors rely on at least two approximations. One is using pseudo-likelihood and the other is using the most likely generation path instead of performing marginalization. It is fine that the authors focus on empirical results of translation quality but then I would like to see more strong and extensive evidence that supports the use of such approximation.


**Experience Assessment:**

I have published in this field for several years.

**Review Assessment: Checking Correctness Of Derivations And Theory:**

I assessed the sensibility of the derivations and theory.

**Review Assessment: Checking Correctness Of Experiments:**

I assessed the sensibility of the experiments.

**Review Assessment: Thoroughness In Paper Reading:**

I read the paper at least twice and used my best judgement in assessing the paper.

---

> ### Author Response · Authors · 2019-11-10
> **Thanks for the review!**
>
> We believe that our framework is a probabilistically sound way of generating from masked language models, and language models generally. Our methodology does require some approximations to tractably generate sentences, but the overall generative story we offer is still valid. Moreover, even generating in traditional left-to-right autoregressive models requires approximations (e.g. greedy decoding or beam search) due to the exponential hypothesis space.
>
> We believe our approximations are theoretically sensible, but they are difficult to verify how good they are due to the fact that what we’re trying to approximate in the first place is highly intractable to compute. One piece of evidence we’ve looked at and included in the paper (Fig 3 in appendix in revision) is the probability of the intermediate sequences over the generation process. The energy (in the sense of energy-based models, see https://arxiv.org/abs/1902.04094) of the target sequence decreases (equivalently, the probability increases) throughout the generation process. The figure also shows that more sophisticated decoding schemes such as left2right, easy-first, and least2most converge to lower energy (higher probability) target sequences faster compared to uniformly picking positions. Overall, we believe that this is evidence that even with the approximations involved, our framework is able to find target sequences in a probabilistically sound manner and points to the value of further research in developing better coordinate selection mechanisms. If the reviewer has any thoughts on further experiments or evidence to explore, we’d be happy to hear it.

---

### Official Review · AnonReviewer1 · 2019-10-25
**Official Blind Review #1**

**Rating:** 3

**Review:**

This paper focuses on decoding/generation in neural sequence models (specifically machine translation) in a non-autoregressive manner that instead of generating in a left-to-right manner, focuses on generating sequences by picking a length first ,and then indices to replace in a deterministic or random scheme and, finally using a context sensitive distribution over vocabulary (using BERT-like masked LM scheme)  to pick the word to replace. In practice, this procedure of picking indices and words to replace is repeated T number of times and hence the final sequence is obtained by this iterative refinement procedure. This is an interesting and important research direction because not only would it result in better and context sensitive greedy/approximate-MAP decoded outputs, but also opens up opportunities for parallelization of the decoding procedure which is difficult to achieve with left-to-right decoders.
That said, the results are fairly inconclusive and the practical implementation does leave things desired for a practical decoder. As observed by the authors, different deterministic strategies for choosing T results in very different performances among the variants of the proposed approach. Besides among the variants, one clear pattern is that uniformly random picking of indices is worse than other schemes (left-to-right, least-to-most, easy-first) which is not unexpected but no conclusive empirical evidence can be found for relative differences between the performances of other 3 schemes. Moreover, the proposed decoding variants generally perform worse than or at best similarly to standard autoregressive baselines. As authors note, this is due to the mismatch between the method in which the model was trained and the decoding procedure which is not surprising, but at the same time this does not give insight into the effectiveness of the proposed decoding objective. The central question is: if the training prefers left-to-right generation then how valuable is it to device  more reasonable but incompatible decoding procedures?

Also, authors also note that index picking schemes investigated in the paper are heuristic based and a more interesting decoder could be learned if index selection procedure itself was learned with features depending on the previous index selection states, decoded states Y, and other relevant quantities. They attribute poor performance of the proposed decoder to the nature of index selection approaches investigated in the paper. I think the paper would be strengthened with results with a more sophisticated learned index selection procedure in addition to the heuristics investigated in this paper.

Overall, while the idea and motivation behind this work is exciting, the inconclusive results and the approaches for practical implementation leave open significant room for improvement.

**Experience Assessment:**

I have published one or two papers in this area.

**Review Assessment: Checking Correctness Of Derivations And Theory:**

I carefully checked the derivations and theory.

**Review Assessment: Checking Correctness Of Experiments:**

I carefully checked the experiments.

**Review Assessment: Thoroughness In Paper Reading:**

I read the paper thoroughly.

---

> ### Author Response · Authors · 2019-11-10
> **Thanks for the review!**
>
> We appreciate that you found our work exciting! We disagree that there is no clear pattern on which decoding strategy to use for linear/constant time decoding scenarios. For linear time case (Table 1), left2right and easy-first decoding strategies outperform both least2most and uniform decodings while left2right having slight performance improvement over easy-first decoding strategy. These results hold across various linear-time decoding hyperparameter settings (beam size, decoding budget (T) and whether using reranking with an autoregressive model). For constant-time decoding results (Table 3) least2most decoding strategy works best when annealing number of generated symbols at each step (L -> 1) while easy-first strategy works best when generating constant L/T number of tokens. In practice if one would to use our method, left2right is recommended for linear-time translation and least2most is recommended for constant-time translation with annealing number of tokens L->1.
>
> > “The central question is: if the training prefers left-to-right generation then how valuable is it to device more reasonable but incompatible decoding procedures?”
>
> We don’t think it’s obvious that left2right decoding is the best decoding strategy for all possible decoding settings. For example, from our experiments in the constant time decoding, left2right decoding performs considerably worse than least2most decoding. Furthermore, given the proliferation of and advantages of non-left-to-right pretraining objectives, we argue that it is worthwhile to investigate non-left-to-right decoding strategies. Such strategies have additional potential benefits, such as being easier to parallelize.
>
> > “I think the paper would be strengthened with results with a more sophisticated learned index selection procedure in addition to the heuristics investigated in this paper.”
>
> We would agree that learning position selecting mechanism would be interesting to investigate, but we think this would be a significant undertaking on its own, as we would need to develop and test ways of providing generation order error signal and methods to learn against that signal.

---

> > ### Comment · AnonReviewer1 · 2019-11-14
> > **Thanks for the response**
> >
> > I thank the authors for the response to the review. The authors point out that under different time-constraints, they observe performance difference between different strategies. But for a general purpose decoder, I still feel that results regarding the best ordering strategy are inconclusive.
> >
> > Moreover, while I understand the importance of working on non left-to-right decoding strategies, my comment was about working on decoding in isolation instead of thinking about both training and decoding since the training/inference compatibility plays a huge role performance-wise.
> >
> > Also, in light of the mixed results, my mind is unchanged on inclusion of learnable position selection mechanisms.

---

> > > ### Author Response · Authors · 2019-11-15
> > > **Thanks for following up**
> > >
> > > Thanks for following up on the review!
> > >
> > > What is your definition of general purpose decoder ? We are hoping to clarify it in order to understand better how decoding time constraints fit in your definition of general purpose decoder.
> > >
> > > With a learnable position selection mechanisms, we would imagine that under different time constraints (linear-time vs constant-time decoding) the model would learn different position selection strategies and time budget is important to include in the error signal.

---

### Official Review · AnonReviewer2 · 2019-10-30
**Official Blind Review #2**

**Rating:** 6

**Review:**

This paper proposes a generalized framework for sequence generation that can be applied to both directed and undirected sequence models. The framework generates the final label sequence through generating a sequence of steps, where each step generates a coordinate sequence and an intermediate label sequence. This procedure is probabilistically modeled by length prediction, coordinate selection and symbol replacement. For inference, instead of the intractable naive approach based on Gibbs sampling to marginalize out all generation paths, the paper proposes a heuristic approach using length-conditioned beam search to generate the most likely final sequence. With the proposed framework, the paper shows that masked language models like BERT, even though they are undirected sequence models, can be used for sequence generation, which obtains close performance to the traditional left-to-right autoregressive models on the task of machine translation.

Overall the paper has significant contributions in the following aspects:
1. It enables undirected sequence models, like BERT, to perform decoding or sequence generation directly, instead of just serving as model pre-training.
2. The proposed framework unifies directed and undirected sequence models decoding, and it can represent a few existing sequence model decoding as special cases.
3. The coordinate sequence selection function in the framework can be dependent on the intermediate label sequence. A few simple selection approaches proposed in the paper are shown to be effective. It could be further extended.
4. The analysis of the coordinate selection order is interesting and helpful for understanding the algorithm.
5. The experiment results for decoding masked language models on machine translation are promising. It also provides the comparison to recent related work on non-autoregressive approaches.

The presentation of the paper is also clear. I am leaning towards accepting the paper.

However, there are some weaknesses:

1. It should be analyzed more why different coordinate selection approaches perform differently in linear-time decoding vs. constant-time decoding. Even in constant-time decoding, the conclusion varies in different decoding setting, easy-first is the worst for the L->1 case, but the best for the L/T case, why is that?

2. What is the motivation for "hard-first"?

3. The setting of "least2most" with L->1 is similar to Ghazvininejad et al. 2019. But Table 4 in the appendix shows the result in this paper is still worse (21.98 vs. 24.61, when both systems use 10 iterations without AR). Also, the gap from the AR baseline is larger than that in Ghazvininejad et al. 2019. Given the two systems are considered similar, it should be explained in the paper the possible reasons for these discrepancies in results.

Additional minor comments for improving the paper:

1. In the introduction, it mentions the baseline AR is (Vaswani et al. 2017), while in the experimental settings, it mentions the baseline AR is (Bahdanau et al. 2015). Please clarify which one is used.

2. In Table 1, how does T = 2L work for the "Uniform" case while the target sequence length is only T, since it is mentioned the positions are sampled without replacement. Similarly, how does T = 2L work for the "Left2Right" case? Is it just always choosing the last position when L < t <= 2L? In these two cases, it seems T > L is not needed.

3. In Table 3, the header for the 2nd column should be o_t, as defined in Section 4 - "Decoding scenarios". What is the actual value of K and K'' for the constant-time machine translation experiments in the paper?

4. "Rescoring adds minimal overhead as it is run in parallel" - it still needs to run left-to-right in sequence since it is auto-regressive. Please clarify what it means by "in parallel" here.

5. What is the range and average for the target sentence length? How is T = 20 for constant-time decoding compared to linear-time decoding in terms of speed?

**Experience Assessment:**

I have read many papers in this area.

**Review Assessment: Checking Correctness Of Derivations And Theory:**

I assessed the sensibility of the derivations and theory.

**Review Assessment: Checking Correctness Of Experiments:**

I carefully checked the experiments.

**Review Assessment: Thoroughness In Paper Reading:**

I read the paper at least twice and used my best judgement in assessing the paper.

---

> ### Author Response · Authors · 2019-11-10
> **Thanks for the review!**
>
> We appreciate that you found our work interesting!
>
> > “It should be analyzed more why different coordinate selection approaches perform differently in linear-time decoding vs. constant-time decoding. Even in constant-time decoding, the conclusion varies in different decoding setting, easy-first is the worst for the L->1 case, but the best for the L/T case, why is that?”
>
> In the L->1 case, we refine tokens multiple times, so it makes sense to generate the hard tokens first and refine them several times, rather than generate the hard tokens at the end and get no opportunity to refine them. In the L->T setting, there is no refinement, so generating the easy tokens first makes sense as it gives additional context for the hard tokens.
>
> > “What is the motivation for "hard-first"?”
>
> As stated above, our intuition is that in settings where we allow the model to refine its predictions, it makes sense for the model to first generate the hard-to-predict tokens and get multiple attempts to fix them, rather than predicting the hard tokens at the end and not getting a chance to refine them.
>
> > “The setting of "least2most" with L->1 is similar to Ghazvininejad et al. 2019. But Table 4 in the appendix shows the result in this paper is still worse (21.98 vs. 24.61, when both systems use 10 iterations without AR). Also, the gap from the AR baseline is larger than that in Ghazvininejad et al. 2019. Given the two systems are considered similar, it should be explained in the paper the possible reasons for these discrepancies in results.”
>
> There are differences in both model and training hyperparameters between our work and work by Ghazvininejad et al. 2019. We use smaller Transformer model with 1024 hidden units vs 2048 units in Ghazvininejad et al. 2019. We also train the model with more than twice smaller batch size since we use 8 GPUs on DGX-1 machine and Ghazvininejad et al. 2019 use 16 GPUs on two DGX-1 machine with float16 precision. Finally we don’t average best 5 checkpoints and don’t use label smoothing for our model.
>
> > “In the introduction, it mentions the baseline AR is (Vaswani et al. 2017), while in the experimental settings, it mentions the baseline AR is (Bahdanau et al. 2015). Please clarify which one is used.”
>
> We used Transformer AR model (Vaswani et al 2017) initialized with the same pretrained model weights used for our model. Bahdanau et al 2015 used a reference to general autoregressive machine translation approach with attention. We removed Bahdanau et al 2015 citation from that paragraph to remove ambiguity in the revision.
>
> > “In Table 1, how does T = 2L work for the "Uniform" case while the target sequence length is only T, since it is mentioned the positions are sampled without replacement. Similarly, how does T = 2L work for the "Left2Right" case? Is it just always choosing the last position when L < t <= 2L? In these two cases, it seems T > L is not needed.”
>
> In uniform case, position selection is done by sampling uniformly without replacement with population of the set of all possible positions for a defined length. During generation once number of generation steps T becomes the same as length L we reset the population and do uniform sampling without replacement again. For left2right case when T = 2L the generation process goes left-to-right twice, resetting at the leftmost token after L steps.
>
> > “In Table 3, the header for the 2nd column should be o_t, as defined in Section 4 - "Decoding scenarios". What is the actual value of K and K'' for the constant-time machine translation experiments in the paper?”
>
> Thanks for pointing out that the header typo! It should be updated in the revision. We don’t do beam search for constant-time machine translation so K=K’’=1.
>
> > “"Rescoring adds minimal overhead as it is run in parallel" - it still needs to run left-to-right in sequence since it is auto-regressive. Please clarify what it means by "in parallel" here.”
>
> For rescoring we only need to obtain log probabilities given the full sequence available to us. Compared to generation where tokens are generated sequentially one at a time in left-to-right order, rescoring is done in parallel where left-to-right constraint is enforced by masking out future tokens (i.e. tokens to the right of current token)
>
> > “What is the range and average for the target sentence length? How is T = 20 for constant-time decoding compared to linear-time decoding in terms of speed?”
>
> For the WMT’14 En-De dataset we used, the average target sentence length for both English and German sides is about 25 and sentence lengths range from 3 up to 120.

---

### Decision · Program_Chairs · 2019-12-19

**Decision:**

Reject

**Comment:**

This paper proposes a generalized way to generate sequences from undirected sequence models.

Overall, I believe a framework like this could definitely be a valuable contribution, but as Reviewer 1 and Reviewer 3 noted, the paper is a bit lacking both in theoretical analysis and strong empirical results. I don't think that this is a bad paper at all, but it feels like the paper needs a little bit of an extra push to tighten up the argumentation and/or results before warranting publication at a premier venue such as ICLR. I'd suggest the authors continue to improve the paper and aim to re-submit at revised version at a future conference.